# Systemic Antibiotic Prophylaxis to Reduce Early Implant Failure: A Systematic Review and Meta-Analysis

**DOI:** 10.3390/antibiotics10060698

**Published:** 2021-06-10

**Authors:** Elisabet Roca-Millan, Albert Estrugo-Devesa, Alexandra Merlos, Enric Jané-Salas, Teresa Vinuesa, José López-López

**Affiliations:** 1Department of Odontostomatology, School of Dentistry, Faculty of Medicine and Health Sciences, University of Barcelona, 08907 Barcelona, Spain; erocamil10@alumnes.ub.edu (E.R.-M.); albertestrugo@ub.edu (A.E.-D.); enricjanesalas@ub.edu (E.J.-S.); 2Department of Pathology and Experimental Therapeutics, School of Dentistry and Medicine, Faculty of Medicine and Health Sciences, University of Barcelona, 08907 Barcelona, Spain; amerlos@ub.edu (A.M.); tvinuesa@ub.edu (T.V.)

**Keywords:** antibiotics, systemic antibiotic prophylaxis, early implant failure, dental implants, systematic review, meta-analysis

## Abstract

Systemic antibiotics are routinely prescribed in implant procedures, but the lack of consensus causes large differences between clinicians regarding antibiotic prophylaxis regimens. The objectives of this systematic review are to assess the need to prescribe antibiotics to prevent early implant failure and find the most appropriate antibiotic prophylaxis regimen. The electronic search was conducted in PubMed/MEDLINE, Scielo and Cochrane Central Trials Database for randomized clinical trials of at least 3 months of follow-up. Eleven studies were included in the qualitative analysis. Antibiotics were found to statistically significantly reduce early implant failures (RR = 0.30, 95% CI: 0.19–0.47, *p* < 0.00001; heterogeneity I^2^ = 0%, *p* = 0.54). No differences were seen between preoperative or both pre- and postoperative antibiotic regimens (RR = 0.57, 95% CI: 0.21–1.55, *p* = 0.27; heterogeneity I^2^ = 0%, *p* = 0.37). A single preoperative antibiotic prophylaxis dose was found to be enough to significantly reduce early implant failures compared to no antibiotic (RR = 0.34, 95% CI: 0.21–0.53, *p* < 0.00001; heterogeneity I^2^ = 0%, *p* = 0.61). In conclusion, in healthy patients a single antibiotic prophylaxis dose is indicated to prevent early implant failure.

## 1. Introduction

The failure of dental implants according to chronological criteria can be divided into early failure, when it occurs before or during the abutment connection, or late failure, after prosthetic loading [1]. Early implant failure is due to the lack of osseointegration [1] and occurs in approximately 2% of cases [2]. It has been linked to numerous factors such as lack of primary stability, contamination of the implant surface during placement, surgical trauma, excessive micromovements during the healing process, poor bone quality or quantity, implants with reduced length or smoking among others [2,3,4,5,6,7].

Due to the diversity of the microbiota present in the oral cavity, the second most diverse and large on the human body [8], systemic antibiotics are routinely prescribed to prevent infection and the consequent early implant failure [9,10]. However, there is no standardized protocol regarding antibiotic prescription and cross-sectional studies have revealed large differences in the type of antibiotic, dosage, time of administration and duration of the treatment [9,10]. Furthermore, the scientific evidence is not solid on the need for systemic antibiotics in oral implantology [11,12] and it must be considered the possible adverse effects related to these drugs [13] and the continuous increase of antibiotic resistance [14].

Randomized clinical trials (RCTs) on this topic show different conclusions: that antibiotic prescription may not be necessary in case of proper asepsis [12] or in single implants [11], that it is justified since it reduces early implant failure and results in a better postoperative period [15], or that significantly reduces early implant failure [16].

Regarding the type of administration, most of the studies suggest that a single preoperative dose of antibiotic is enough to prevent the early implant failure, thus avoiding long-term antibiotic administration [16,17,18,19]. Even so, clinicians tend to prescribe long duration antibiotic regimens, pre- and postoperative administration are the most common [10].

The type of antibiotic is the only aspect in which there seems to be a consensus, amoxicillin being the most widely prescribed antibiotic in implant dentistry followed by amoxicillin in association with clavulanic acid [9,10]. Likewise, clindamycin is the most common alternative in patients allergic to penicillin [9].

Due to this lack of consensus and with the aim of shedding light on this issue, the objectives of this systematic review are to assess the need for systemic antibiotics in patients undergoing dental implant placement to prevent early implant failures and to find the most appropriate antibiotic prophylaxis regimen in these procedures considering its efficacy and the prescribed dose.

## 2. Materials and Methods

The present systematic review and meta-analysis was performed in accordance with the Preferred Reporting Items of Systematic Reviews and Meta-Analyses (PRISMA) statement [20]. A detailed protocol was prepared before starting the review and registered on Prospero (CRD42021248348).

### 2.1. Focused Questions

Does the administration of antibiotic prophylaxis reduce early implant failure?

If the answer is yes, what is the best antibiotic prophylaxis regimen to reduce early implant loss?

### 2.2. PICO Question

P (population): Partial or totally edentulous patients undergoing dental implant surgery.

I (intervention): Administration of any type of systemic antibiotic, regardless of dosage, duration or time of administration.

C (comparation): Different type of antibiotic, dosage, duration or time of administration, or compared to placebo or no antibiotic.

O (outcome): Early implant failure (dental implants that must be removed before prosthetic loading because of mobility or infection).

### 2.3. Eligibility Criteria

Inclusion criteria:Randomized Clinical Trials.At least 3 months follow-up.Antibiotic prophylaxis regimen clearly described.Partial or totally edentulous patients undergoing dental implant surgery.

Exclusion criteria:Local antibiotic application instead of systemic antibiotic prophylaxis.Patients requiring antibiotic prophylaxis for medical reasons.

### 2.4. Search Strategy

An electronic search of the literature was conducted by two authors (E.R.-M. and A.E.-D.) for articles written in English or Spanish published from January 2000 to April 2021. The consulted databases were PubMed/MEDLINE, Scielo and Cochrane Central Trials Database. An additional hand search was performed to identify studies with potential of inclusion in the references of the articles identified in the electronic search. Both searches were conducted on 9 April 2021. The following term combination was used in all databases: (“antibiotics [All Fields]” OR “anti-bacterial agents [All Fields]” OR “amoxicillin [All Fields]”) AND (“dental implant [All Fields]” OR “implantology [All Fields]” OR “implant placement [All Fields]” OR “implant failure [All Fields]”).

### 2.5. Study Selection

The study selection was carried out by two authors (E.R.-M. and A.E.-D.). After discarding duplicate articles, the titles were read to identify studies with potential for inclusion. Then, the abstracts were read to discard papers that did not meet the inclusion/exclusion criteria. Full text of the selected articles was read to verify that they met the inclusion/exclusion criteria. Disagreements between the two authors were solved consulting a third author (J.L.-L.).

### 2.6. Data Extraction

The data were independently extracted by two authors (E.R.-M. and A.M.) and entered in data collection forms (Microsoft Excel version 16.35). In case of disagreement a third author was consulted (J.L.-L.). The collected data included author(s), year of publication, country, type and number of centers, follow-up period, number of patients and implants analyzed, inclusion and exclusion criteria, aseptic measures before and after surgery and type of antibiotic, dosage, duration and time of administration of the different test and control groups. Corresponding authors were contacted in the event that data were missing.

### 2.7. Risk of Bias in Included Studies

Version 2 of the Cochrane Collaboration’s tool for assessing risk of bias in randomized clinical trials (RoB 2) [21] was implemented to assess the risk of bias of the RCT. The evaluated domains were: randomization process, deviations from intended interventions, missing outcome data, measurement of the outcome and selection of the reported result. Studies were classified as: “low risk of bias”, “high risk of bias” or “some concerns”. Robvis web app was used to create the risk of bias plots [22].

### 2.8. Data synthesis and Statistical Analysis

A qualitative analysis of the included articles was be carried out by grouping those according to the characteristics of the test and control groups.

For the quantitative analysis, pooled estimates from the studies were performed using a fixed or random effect-model depending on the heterogeneity (I^2^ ≥ 40%). Review Manager (RevMan) (computer program, version 5.4, The Cochrane Collaboration, 2020) was implemented to calculate the Risk Ratio (RR) and 95% confidence interval. The level of significance was set at *p* < 0.05 and heterogeneity was evaluated with Chi^2^ and I^2^ tests. The statistical unit was the dental implant.

## 3. Results

### 3.1. Study Selection

Through the electronic search, a total of 749 records were identified. After discarding the duplicates, 710 titles and, if necessary the abstracts, were read to assess the potential for inclusion. A total of 698 studies were excluded. The full text of the remaining 12 articles and the two works identified through hand search were evaluated to verify they met the inclusion/exclusion criteria [11,12,15,16,17,18,19,23,24,25,26,27,28,29]. One was discarded because it was a quasi-random controlled clinical trial [25], a second one was excluded because the assessed variable was not early implant failure [24] and another was ruled out because the follow-up period was 8 weeks [23]. Finally, 11 studies were included in the qualitative analysis [11,12,15,16,17,18,19,26,27,28,29] (Figure 1).

### 3.2. Study Methods and Characteristics

All the included studies were RCT, published between 2008 and 2019 (Table 1). Five of them were carried out in a single center (a total of 327 patients, minimum 46 and maximum 80, and 624 implants, minimum 46 and maximum 247) [12,15,17,18,29] and the other six were multicenter (1817 patients, minimum 100 and maximum 506, and 3413 implants, minimum 105 and maximum 972) [11,16,19,26,27,28]. Four took place in university centers (247 patients, minimum 46 and maximum 80, and 534 implants, minimum 46 and maximum 247) [12,15,17,29] and the other seven in private practice clinics (1897 patients, minimum 80 and maximum 506, and 3503 implants, minimum 90 and maximum 972) [11,16,18,19,26,27,28]. The mean follow-up period was 4.5 months (range 3–10). The total number of patients and implants analyzed were 2144 and 4037, respectively. The inclusion and exclusion criteria of the included studies and the hygienic and asepsis measures pre- and post-surgery are collected in Table 2.

Five RCT compared antibiotic prophylaxis regimens with placebo administration (1028 patients, minimum 46 and maximum 506, and 1874 implants, minimum 46 and maximum 972) [11,15,27,28,29]. Another three compared antibiotic prophylaxis administration with no antibiotic (627 patients, minimum 80 and maximum 447, and 1358 implants, minimum 148 and maximum 963) [12,16,26]. The last three compared different antibiotic prophylaxis regimens without a placebo or no antibiotic group (489 patients, minimum 66 and maximum 343, and 805 implants, minimum 90 and maximum 529) [17,18,19]. The antibiotic used was in all cases amoxicillin, except in one study in which allergic to penicillin patients were also included and were prescribed clindamycin [16].

Most studies had a preoperative antibiotic prophylaxis group (978 patients, minimum 25 and maximum 252, and 1853 implants, minimum 27 and maximum 535) [11,15,16,17,18,19,26,27,28], five studies had a group with antibiotic administration pre- and post-surgery (313 patients, minimum 25 and maximum 177, and 595 implants, minimum 36 and maximum 128) [12,17,18,19,26] and only two RCT had a group which only was prescribed postoperative antibiotic regimen (48 patients, minimum 23 and maximum 25, and 72 implants, minimum 24 and maximum 48) [26,29] (Table 3). One RCT did not specify the number of implants per group [17].

In cases in which a single dose antibiotic prophylaxis was prescribed, the most frequent dose was 2 g amoxicillin 1 h before the surgery. Although in one study 1 g amoxicillin was administered [18] and in another 3 g amoxicillin [15]. In case of patients allergic to penicillin, 600 mg clindamycin was administered 1 h before the implant placement [16].

In cases in which only postoperative antibiotic was administered, the duration of the treatment was 7 days [26,29]. When pre- and postoperative antibiotic regimens were prescribed, only two of the studies administered 7 days of antibiotic post-surgery [17,26], the other three opted for short term antibiotic regimens (2–3 days) [12,18,19].

Regarding hygienic and aseptic measures, in five of the studies patients received prophylaxis, debridement and/or oral hygiene instructions when required a few days before implant surgery [11,26,27,28,29]. In most of the RCT, chlorhexidine mouthwash rinses were performed before [11,15,17,18,19,26,27,29] or after the intervention [15,18,19,26,27,28,29].

Regarding the complexity, two studies specified that the surgical procedures involved only the placement of simple single dental implants [11,29]. Data about the inclusion/exclusion of immediate implants, implants with immediate loading and implants with simultaneous bone grafting in the included studies are summarized in Table 2.

### 3.3. Risk of Bias in Included Studies

Four of the included RCT obtained a low overall risk of bias [11,18,27,28] while in the others there were some concerns regarding the overall risk [12,15,16,17,19,26,29]. The domain with the highest risk of bias was the one referring to the deviations from intended interventions. The domains in which a lower risk of bias was obtained were those that refer to missing outcome data and measurement of the outcome (Figure 2).

### 3.4. Early Implant Failure

Of the eight studies that compared the administration of antibiotic prophylaxis with no antibiotic or with placebo [11,12,15,16,26,27,28,29], only one found statistically significant differences between groups [16]. That RCT compared the preoperative administration of amoxicillin 2 g or clindamycin 600 mg with no antibiotic. Even so, the rest of the studies found less early implant failures in the antibiotic groups [11,12,15,16,26,27,28,29].

Four RCT compared preoperative antibiotic regimen with both pre- and postoperative antibiotic regimen, and in none of the cases were statistically significant differences found [17,18,19,26]. Only one work compared a postoperative antibiotic regimen with preoperative and both pre- and postoperative regimens and observed no statistically significant differences [26].

The implant failure rate in the antibiotic groups was 1.55%, while in the no antibiotic or placebo groups was 4.61%.

### 3.5. Meta-Analyses

Different pooled estimates were performed using a fixed effect-model because heterogeneity was I^2^ < 40%. The prescription of antibiotic prophylaxis, regardless of the dosage or the time of administration, proved to be a protective factor for early implant failure, with statistically significant results (RR = 0.30, 95% CI: 0.19–0.47, *p* < 0.00001; heterogeneity I^2^ = 0%, *p* = 0.54) [11,12,15,16,26,27,28,29] (Figure 3). 

The meta-analysis comparing preoperative antibiotics versus pre- and postoperative antibiotics did not show differences between groups (RR = 0.57, 95% CI: 0.21–1.55, *p* = 0.27; heterogeneity I^2^ = 0%, *p* = 0.37) [17,18,19,26]. It must be considered that in two of the three studies included there was no implant failure in either of the two groups [18,26], so the effect of the different antibiotic regimens in these two studies was not estimable. For this reason, the result depended only on two studies [17,19] (Figure 4).

A single preoperative antibiotic dose compared with no antibiotic prescription, turned out to be a significant protective factor in early implant failure (RR = 0.34, 95% CI: 0.21–0.53, *p* < 0.00001; heterogeneity I^2^ = 0%, *p* = 0.61) [11,15,16,26,27,28] (Figure 5).

## 4. Discussion

According to the results obtained in the present systematic review and meta-analysis, antibiotic prophylaxis is indicated in dental implant procedures since they statistically significantly reduce early implant failure. No difference has been observed between the administration of a single preoperative antibiotic dose and the combination of pre- and postoperative antibiotics. A single preoperative dose has been found to be enough to significantly reduce dental implant failures compared to no antibiotic or placebo.

Even so, the most recent cross-sectional studies reveal an antibiotic over prescription in relation to this type of surgical procedures, being very common in some countries, such as Spain and Italy, the prescription of antibiotic from one or two days before implant placement to five or seven days after surgery [9,10].

The results obtained agree with those found in recent systematic reviews and meta-analyzes [30,31,32,33]. Canullo et al. concluded that antibiotic prophylaxis prevents early implant failures in healthy patients [31]. Romandini et al. determined that antibiotic prophylaxis is necessary in implant placement procedures but that there is insufficient evidence to recommend a specific dosage [30]. They also concluded that the prescription of postoperative antibiotics is not justified [30]. Finally, Kim et al. affirmed that the administration of antibiotic prophylaxis reduces the risk of early implant failure but further research is necessary to stablish a standardized protocol [32].

It should be taken into account that the included RCTs treated patients without systemic conditions that could pose a greater risk of implant failure, so if the protective effect of the antibiotic is observed in healthy patients, its use is indisputable in patients at risk.

All the studies that focused on the effect of antibiotic regimens in early implant failure used amoxicillin as the antibiotic of choice or, in case of allergy, clindamycin. However, it appears that bacteria genes encoding resistance to metronidazole are less frequent in peri-implant tissues than those that encode resistance to beta-lactams [34]. In addition, bacteria related to dental implant failure [35] and peri-implantitis [36,37] can be variable and complex, but it consists in part of Gram-negative anaerobic species against which metronidazole is the antibiotic of choice [38,39]. For these reasons, it would be interesting to investigate the effectiveness of metronidazole in reducing dental implant failure.

On the other hand, local antibacterial drug delivery systems are being investigated in the field of oral implantology, which would allow to maintain the minimum inhibitory concentration of antibiotic around dental implants during the osseointegration time [40,41].

Although the mechanisms of antibiotic therapy in implant dentistry could change a lot in the coming decades, due to the large number of dental implants placed each year and therefore the large number of patients undergoing antibiotic prophylaxis, it is essential to establish antibiotic protocols both for patients at risk and for healthy patients. It would also be interesting to differentiate between simple and complex implant procedures.

The limitations of the present study are the small sample and the unclear risk of bias especially regarding the blinding of participants and researchers in some of the included studies, the low number of RCTs with a follow-up period of at least 3 months that analyze this question and the lack of studies comparing different doses of preoperative amoxicillin prophylaxis.

## 5. Conclusions

Systemic antibiotic prophylaxis significantly reduces early dental implant failures in healthy patients. There is no statistically significant difference in the occurrence of this event when comparing the pre and postoperative administration of systemic antibiotics with the prescription of a single preoperative antibiotic dose. Therefore, the continuation of antibiotic treatment postoperatively would not be indicated in these patients.

Future research should be aimed at establishing differences between the systemic antibiotic needs of healthy patients versus patients with systemic diseases, of patients with or without different types of oral diseases or comparing simple implant surgeries versus complex procedures. In addition, RCTs comparing different single preoperative antibiotic doses should be performed.

## Figures and Tables

**Figure 1 antibiotics-10-00698-f001:**
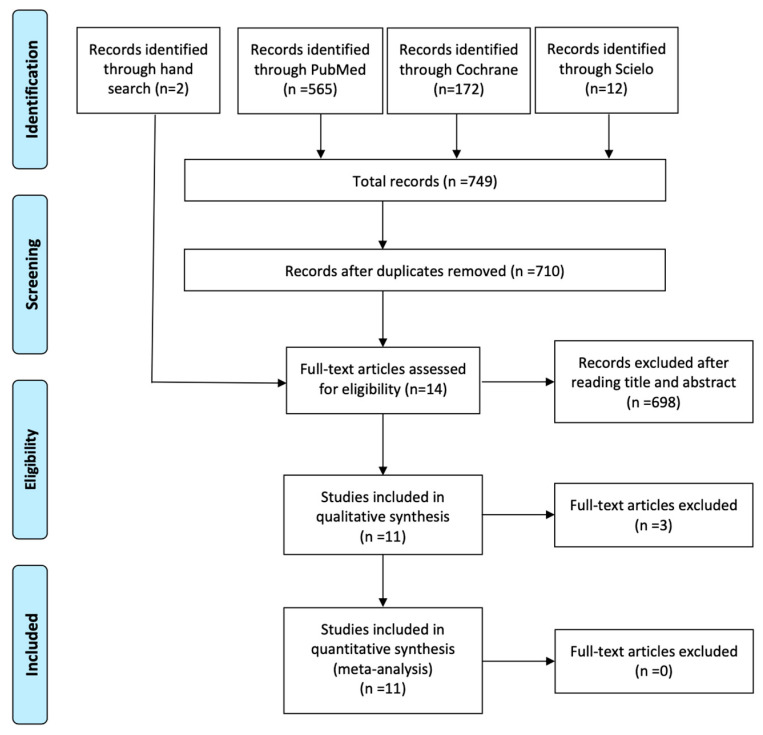
Preferred Reporting Items for Systematic Reviews and Meta-Analyses (PRISMA) flow diagram of the selection process.

**Figure 2 antibiotics-10-00698-f002:**
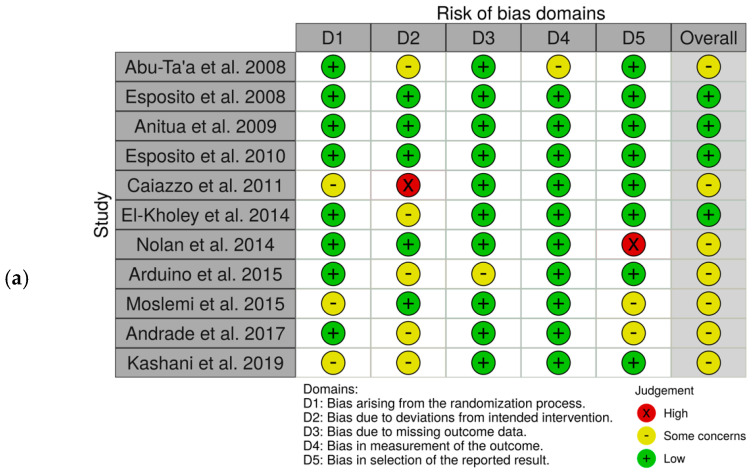
Risk of bias within the included studies. (**a**) Traffic light plots of the domain-level judgements for each individual study; (**b**) weighted bar plots of the distribution of the risk of bias judgement for each domain.

**Figure 3 antibiotics-10-00698-f003:**
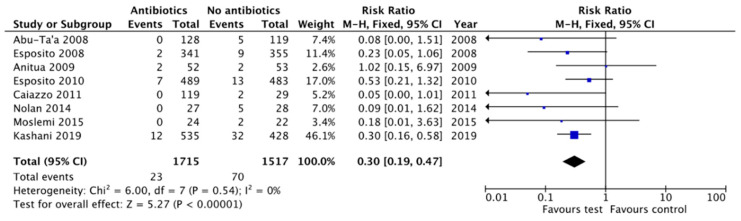
Forest plot for the effect of antibiotic prophylaxis versus no antibiotic or placebo on early implant failure.

**Figure 4 antibiotics-10-00698-f004:**
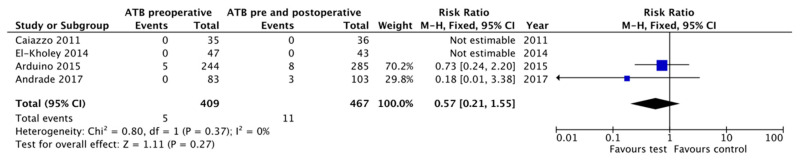
Forest plot for the effect of preoperative antibiotic prophylaxis versus both pre- and postoperative antibiotic prophylaxis on early implant failure.

**Figure 5 antibiotics-10-00698-f005:**
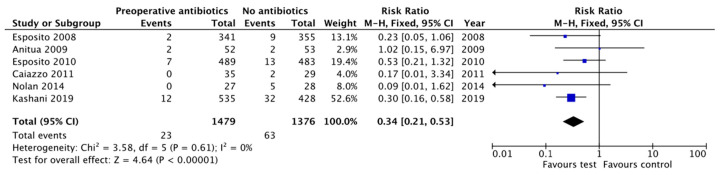
Forest plot for the effect of preoperative antibiotic prophylaxis versus no antibiotic or placebo on early implant failure.

**Table 1 antibiotics-10-00698-t001:** Characteristics of the included studies.

Author	Country	Type of Centre	*n* of Centers	Follow-Up (Months)	Analyzed Patients	Analyzed Implants
Abu-Ta’a et al. 2008 [12]	Belgium	University	Single	5	80	247
Esposito et al. 2008 [28]	Italy	Private practice	Multicentric	4	316	696
Anitua et al. 2009 [11]	Spain	Private practice	Multicentric	3	105	105
Esposito et al. 2010 [27]	Italy	Private practice	Multicentric	4	506	972
Caiazzo et al. 2011 [26]	Italy	Private practice	Multicentric	3	100	148
El-Kholey et al. 2014 [18]	Saudi Arabia	Private practice	Single	3	80	90
Nolan et al. 2014 [15]	Ireland	University	Single	4	55	55
Arduino et al. 2015 [19]	Italy	Private practice	Multicentric	10	343	529
Moslemi et al. 2015 [29]	Iran	University	Single	6	46	46
Andrade et al. 2017 [17]	Brazil	University	Single	3	66	186
Kashani et al. 2019 [16]	Sweden	Private practice	Multicentric	4	447	963

**Table 2 antibiotics-10-00698-t002:** Inclusion/exclusion criteria and hygienic and aseptic measures of the included RCT.

Author	Allergic to Penicillin	Immunosupressed Patients	Endocarditis Prophylaxis	Diabetes Mellitus	Biphosphonates	Previous Head and Neck Radiation	Smokers	Periodontal Patients	Immediate Implants	Immediate Loading	Bone Grafting	Hygiene before Surgery	CHX Pre	CHX Post
Abu-Ta’a et al. 2008 [12]	No	No	No	Not uncontrolled	NA	No	Yes	Yes	NA	NA	NA	NA	NA	NA
Esposito et al. 2008 [28]	No	No	No	No	No (intravenous)	No	Yes	NA	Yes	Yes	No	Yes	NA	Yes
Anitua et al. 2009 [11]	No	No	No	NA	No	<5000 rads	Yes	Yes	NA	Yes	NA	Yes	Yes	NA
Esposito et al. 2010 [27]	No	No	No	No	No (intravenous)	No	Yes	NA	Yes	Yes	No	Yes	Yes	Yes
Caiazzo et al. 2011 [26]	No	NA	No	NA	No	NA	Yes	Yes	NA	NA	NA	Yes	Yes	Yes
El-Kholey et al. 2014 [18]	No	No	No	No	No	NA	Yes	Yes	No	No	No	NA	Yes	Yes
Nolan et al. 2014 [15]	No	No	No	No	NA	No	Yes	No	No	No	No	NA	Yes	Yes
Arduino et al. 2015 [19]	No	No	No	No	No	No	Yes	Yes	NA	No	No	NA	Yes	Yes
Moslemi et al. 2015 [29]	No	No	No	No	No	No	No	No	NA	No	No	Yes	Yes	Yes
Andrade et al. 2017 [17]	No	No	No	No	No	No	Yes	Yes	NA	No	NA	NA	Yes	NA
Kashani et al. 2019 [16]	Yes	No	No	NA	No	No	Yes	Yes	NA	NA	Yes	NA	NA	NA

NA: not available; CHX: chlorhexidine.

**Table 3 antibiotics-10-00698-t003:** Antibiotic regimens within the studies.

Author	Test Group			Control Group
	Preoperative	Postoperative	Pre and Post	
Abu-Ta’a et al. 2008 [12]	-	-	Amoxicillin 1 g 1 h pre and 500 mg × 4 × 2 post	No antibiotics
Esposito et al. 2008 [28]	Amoxicillin 2 g 1 h pre			Placebo
Anitua et al. 2009 [11]	Amoxicillin 2 g 1 h pre	-	-	Placebo
Esposito et al. 2010 [27]	Amoxicillin 2 g 1 h pre			Placebo
Caiazzo et al. 2011 [26]	Amoxicillin 2 g 1 h pre	Amoxicillin 1 g × 2 × 7	Amoxicillin 2 g 1 h pre and 1 g × 2 × 7 post	No antibiotics
El-Kholey et al. 2014 [18]	Amoxicillin 1 g 1 h pre		Amoxicillin 1 g 1 h pre and 500 mg × 3 × 3 post	
Nolan et al. 2014 [15]	Amoxicillin 3 g 1 h pre	-	-	Placebo
Arduino et al. 2015 [19]	Amoxicillin 2 g 1 h pre		Amoxicillin 2 g 1 h pre and 1 g × 2 × 3 post	
Moslemi et al. 2015 [29]	-	Amoxicillin 500 mg × 3 × 7	-	Placebo
Andrade et al. 2017 [17]	Amoxicillin 2 g 1 h pre		Amoxicillin 2 g 1 h pre and 500 mg × 3 × 7 post	
Kashani et al. 2019 [16]	Amoxicillin 2 g or clindamycin 600 mg 1 h pre	-	-	No antibiotics

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
