# Peer review of "Systemic Antibiotic Prophylaxis to Reduce Early Implant Failure: A Systematic Review and Meta-Analysis"

_antibiotics, 2021, doi:10.3390/antibiotics10060698_

Round 1

Reviewer 1 Report

The present systematic review and meta-analysis focused on assessing the need to prescribe antibiotics to prevent early implant failure and finding the most appropriate antibiotic prophylaxis regimen. Eleven studies were included in the qualitative analysis. Antibiotics were found to statistically significantly reduce early implant failures. In conclusion, in healthy patients a single antibiotic prophylaxis dose is indicated to prevent early implant failure.

How about the patients with periodontal disease or other oral diseases? The author should consider these conditions. And the authors need to consider the anti-fungi agents or peri-implantitis as the search term.

Reviewer 2 Report

The topic of this systematic review and meta-analysis may be interesting in a wider perspective on implant procedures, especially during adulthood.

Title

- The title should be more informative, I suggest modifying it with this sentence: “Systemic antibiotic prophylaxis to reduce the early implant failure: a systematic review and meta-analysis” 

Keywords

- “antibiotic prophylaxis” should be changed. I suggest “systemic antibiotic prophylaxis”.

Introduction

-The introduction is well developed and highlights the intentions of the paper. Please define the aim of the review accurately at the end of this section.

Materials and Methods

- Please enter the registration on the portal Prospero to code your revision.

- Inclusion criteria:Referring to the “PICO question”, you should add also “Partial or totally edentulous patients undergoing dental implant surgery”.

Discussions

- The discussion section is well developed, often there are repetitions of results. Please review the text and make it more concise. Please, clearly define the chapter of the conclusions, highlighting in a better way what emerged from the review.

Language

- The English language may be improved, thus a native speaker should revise the manuscript before resubmission.
